# Effects of Ridge Tillage and Straw Mulching on Cultivation the Fresh Faba Beans

**Bo Li** [1,2]**, Xinyu Chen** [1]**, Xiaoxu Shi** [2]**, Jian Liu** [2]**, Yafeng Wei** [2,*] **and Fei Xiong** [1,*]

[1] Jiangsu Key Laboratory of Crop Genetics and Physiology, College of Horticulture and Plant Protection, Yangzhou University, Yangzhou 225009, China; libo1226520@163.com (B.L.); dx120190156@yzu.edu.cn (X.C.)

[2] Jiangsu Yanjiang Institute of Agricultural Sciences, Nantong 226541, China; 20182003@jaas.ac.cn (X.S.); ntliuj@sina.com (J.L.)

**\*** Correspondence: w-yafeng@163.com (Y.W.); feixiong@yzu.edu.cn (F.X.)

**Abstract:** Ridge tillage is an effective agronomic practice and a miniature precision agriculture; however, its effects on the growth of faba beans (*Vicia faba* L.) are poorly understood. This study aimed to determine the effect of ridge tillage and straw mulching on the root growth, nutrient accumulation and yield of faba beans. Field experiments were conducted during 2016 and 2017 cropping seasons and comprised four treatments: ridge tillage without any mulching (RT), flat tillage without any mulch (FT), flat tillage with rice straw mulched on the ridge tillage (FTRSM) and ridge tillage with rice straw mulched on the ridge tillage (RTRSM). The RT and RTRSM increased soil temperature and decreased soil humidity and improved soil total nitrogen, total phosphorus, available potassium and organic matter. RT and RTRSM increased the root length density, root surface area, root diameter and root activity of faba beans at flowering and harvest periods. The RT and RTRSM also increased the nitrogen, phosphorus, potassium absorption and the yield of faba beans. These results indicated that ridge tillage and straw mulching affect faba bean growth by improving soil moisture conditions and providing good air permeability and effective soil nutrition supply. This study provides a theoretical basis for the high yield cultivation improvement of faba beans.

**Keywords:** ridge tillage; faba bean; nutrient; growth; yield

## 1. Introduction

Faba beans have been cultivated for 800–1000 years and are an important winter crop in warm temperate and subtropical areas and an essential source of protein rich food in developing countries consumed as a vegetable [1], green or dried, fresh or canned [2,3]. The amount of cultivated faba beans worldwide was estimated to be 2.4 million hectares in 2016 and majority of the total global area of production is located in China, followed by Ethiopia and the European Union; however, the faba bean yield in China was lower than the world average yield [4]. Fresh faba beans have become one of the indispensable bean vegetable varieties in the market of southeast coast and Yangtze River basin in China and have unique roles of improving the multiple cropping index, enriching multiple compound and efficient planting types and increasing farmers' income and agricultural efficiency.

Faba beans are also grown for green manure and can substantially enhance the yields of cereals or other crops [5]. As a friendly former crop of rice, corn, potato and rapeseed, these beans play an important role in rotation or intercropping, especially in fertilizing the soil, protecting farmland ecological environment, reducing weight and agricultural non-point source pollution, increasing efficiency and improving the quality of agricultural products. In China, faba beans are autumn-sown after rice (*Oryza sativa* L.) or intercropped with cotton or maize in southern and western provinces [6]. During production, faba beans often have a low yield and seedling rate, rotting roots and weak seedlings.

Ridge tillage has attracted interest since the early 1980s. This practice effectively reduces soil erosion and increases crop yield [7–10]. The advantages of ridge tillage include improved

soil fertility, water and pest management, deceased greenhouse gas emission [11], increased active soil depth [12–14], increased soil organic carbon (SOC) [15–17] and improved soil temperature and moisture environment for seed germination in early spring [18,19]. Four basic forms have been developed with the needs of crop practical production and the conditions of local climate (Figure 1) [20].

Returning the rice residue is an effective technique to conserve soil productivity [21], provides nutrients and increases chlorophyll and photosynthesis rate in the late growing period [22]. Returning straw to soil can improve soil structure, organic matter content and soil fertility and, thus, provide a good soil environment for crop growth [23]. Studies on the effect of ridge tillage and crop residues on faba bean grain development, specifically root morphology and nutritional element absorption, remain insufficient. In this work, the effect of ridge tillage and straw mulching on soil environment and the root morphology and nutritional element absorption of faba beans were investigated. This study lays a foundation for the high yield cultivation and quality improvement of faba beans and provides a solution to the problem of low yield and rotting roots in rice-faba bean rotation, which is of great importance to the popularization and application of rice-faba bean planting mode.

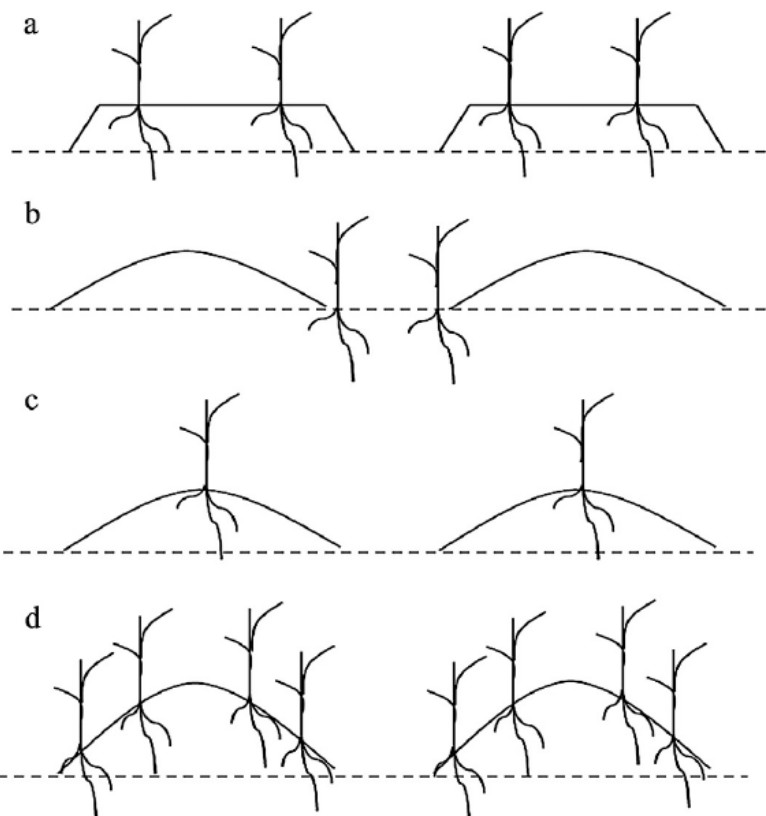

**Figure 1.** Four basic forms in the ridge tillage technology. (**a**–**d**) were bed (wide ridge) cultivation model, ridge and furrow planting of rainfall harvesting model, ridge mulching and planting of warming and water conservation model.

## 2. Materials and Methods

### 2.1. Plant Material and Experimental Treatments

The faba bean cultivar Tongcanxian 7 was provided by Jiangsu Yanjiang Institute of Agricultural Sciences (Nantong, China). A field experiment was conducted from October 2016 to June 2017 in the experimental field of Jiangsu Yanjiang Institute of Agricultural Sciences. The following four treatments were conducted: ridge tillage without any mulching (RT), ridge tillage with rice straw mulched on the surface of ridge (RTRSM), flat tillage

without any mulch (FT) and flat tillage with rice straw mulch (FTRSM). In ridge tillage, the ridge height was 30 cm and the ridge width was 40 cm (Figure 2). The rice straw collected from the rice variety Nanjing 5055 was cut into 10 cm segments, which were planted on 15 May 2016 and mechanically harvested on 20 October 2016. The base fertilizer was applied with 375 kg ha$^{-1}$ compound fertilizer [CO(NH$_2$)$_2$: P$_2$O$_5$: K$_2$O = 15:15:15] and 375 kg ha$^{-1}$ phosphorus fertilizer (P$_2$O$_5$ content = 12%). The planting density of faba beans was 90 thousand plants per hectare. The length of plot was 7.2 m and the width of plot was 2.7 m. Each treatment was replicated three times.

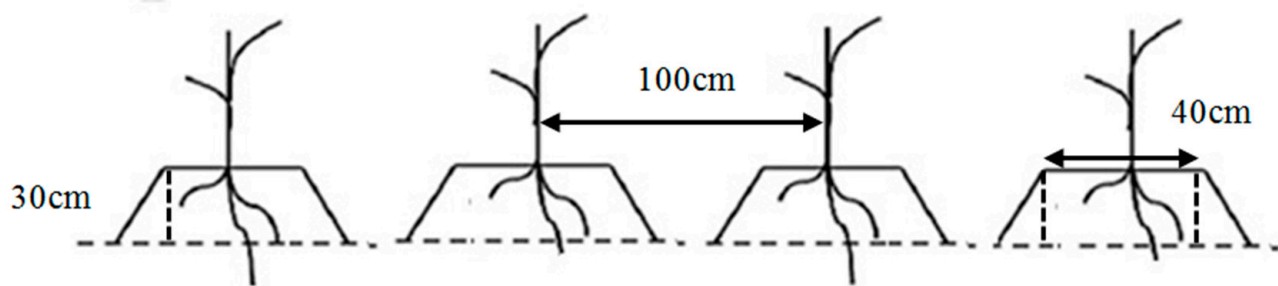

**Figure 2.** Schematic diagram of ridge tillage.

### 2.2. Soil Parameter Determination

Soil temperature and humidity recorder was used to record the soil temperature and humidity in the test plot. Soil samples were collected with a spade from the 0–15 cm depth to analyze total nitrogen, phosphorus, potassium and labile organic matter contents. Total nitrogen content was determined using the Kjeldhal method [24]. Soil samples were first digested with a mixed acid solution of H$_2$SO$_4$ and HClO$_4$ before molybdenum blue method was applied with a spectrophotometer to measure the total phosphorus content [25]. Available potassium content of the soil was obtained by successive extractions with Mehlich-1 extractor (HCl 0.05 M + H$_2$SO$_4$ 0.0125 M). This procedure consisted of conditioning 1.0 g of air-dried soil into Falcon tubes, in which 10 cm$^{-3}$ of Mehlich-1 extractant solution was added. The samples were shaken for five minutes on a horizontal shaker and then centrifuged for five minutes at 2500 RPM. The supernatant was placed in plastic beaker and the potassium content in the extract was determined by flame emission spectrophotometer [26]. Labile organic matter content was analyzed using KMnO$_4$ oxidation method [27].

### 2.3. Root Morphology Observation

Soil blocks of 15 cm × 15 cm × 15 cm were dug and washed with flowing water and the roots were then washed clean with agricultural compression sprayer. Root morphology was scanned using a root scanner (Microtek, scanmaker i800plus, made by Shanghai Zhongjing Technology Co., Ltd., Shanghai, China) and root length density, surface area and average diameter were analyzed by plant root analyzer system (LA-S, made by Hangzhou Wanshen Testing Technology Co., Ltd., Hangzhou, China). The number and dry weight of root nodule per plant were measured.

### 2.4. Root Activity Determination

Root activity was analyzed by the triphenyl tetrazolium chloride (TTC) method [28]. TTC is a chemical that is reduced by dehydrogenases, mainly succinate dehydrogenase, when added to a tissue. Dehydrogenase activity is regarded as an index of the root activity. In brief, 0.5 g of fresh root was immersed in 10 cm$^3$ of equally mixed solution of 0.4% TTC and phosphate buffer and then kept in the dark at 37 °C for 2 h. Subsequently, 2 cm$^3$ of 1 M H$_2$SO$_4$ was added to stop the reaction with the root. The root was dried with filter paper and then extracted with ethyl acetate. The red extractant was transferred into the volumetric flask to reach 10 cm$^3$ by adding ethyl acetate. The absorbance of the extract at

485 nm was recorded. Root activity was expressed as TTC reduction intensity. Root activity = amount of TTC reduction (µg)/fresh root weight (g) × time (h).

### 2.5. Plant Dry Matter and Nutrient Substance Determination

Six representative faba bean plants were taken from each plot at flowering and harvesting periods. The plants were divided into root, stem, leaf and seed. The samples were dried at 105 °C for 30 min and then dried to constant weight at 80 °C. The dry weights of root, stem, leaf and seed were determined and the samples were then ground into powder to determine nitrogen, phosphorus and potassium contents. Nitrogen content was measured by micro-Kjeldahl digestion method, phosphorus content was analyzed colorimetrically by the vanado-molybdate method and potassium content was examined using a flame photometer [29]. The nutrient accumulation of different organs is equal to the nutrient content multiplied by the organ dry weight.

### 2.6. Yield Characteristic Analysis

Fresh faba bean pods were harvested and the yield in each plot was counted at harvesting period. Five representative faba bean plants were taken from each plot at harvesting period. Plant height, branch number of main stem, node number of main stem, pod weight per plant, pod number per plant and 100-seed weight per plant were statistically analyzed.

### 2.7. Statistical Analysis

Data were shown as the mean of triplicate determinations. One-way ANOVA and subsequent least-significant difference (LSD) test were performed using SPSS statistical procedure (SPSS Incorporated, Chicago, IL, USA). Differences at $0.01 \leq p < 0.05$ were considered statistically significant.

## 3. Results

### 3.1. Effect of Ridge Tillage and Straw Mulching on Soil Environment and Nutrient

The soil temperature of RTRSM was the highest (9.07 °C) and was 2.45% higher than that of RT (Figure 3). The soil temperature of FTRSM was 0.47% higher than that of FT, that of RT was 2.91% higher than that of FT and that of RTRSM was 4.98% higher than that of FTRSM. This result indicates that ridge tillage and straw mulched help increase the soil temperature. The soil humidity of FTRSM was higher than that of RTRSM and that of RT was significantly reduced than that of FT. This finding indicates that ridge cover is helpful in reducing soil humidity.

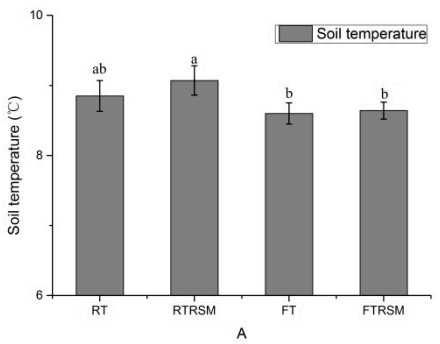
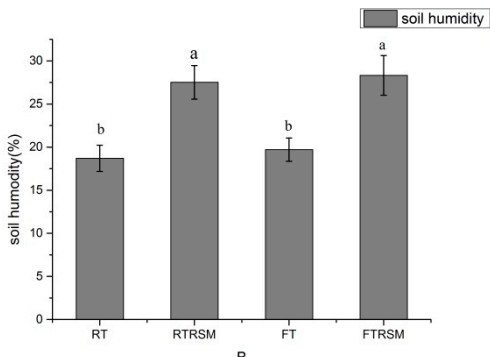

**Figure 3.** Soil temperature and soil humidity under different treatments. (**A**), soil temperature under different treatments. (**B**), soil humidity under different treatments. FT. flat tillage without any mulch; FTRSM, flat tillage with rice straw mulched; RT. ridge tillage without any mulching; RTRSM, ridge tillage with rice straw mulched. Different lowercase above histogram indicates significant difference between treatments.

RT significantly improved the soil total nitrogen content compared with FT. Meanwhile, the soil total nitrogen, phosphorus, soluble potassium and organic matter contents under RTRSM were higher than those under RT. This tendency was consistent with the difference between FTRSM and FT (Table 1). These results indicated that ridge tillage and straw mulching improve the contents of nutrient and organic matter in soil.

**Table 1.** Total nitrogen, phosphorus, available potassium and labile organic matter contents of soil under different treatments.

| Treatment | Total Nitrogen (g kg$^{-1}$) | Total Phosphorus (g kg$^{-1}$) | Available Potassium (mg kg$^{-1}$) | Labile Organic Matter (mg kg$^{-1}$) |
|---|---|---|---|---|
| RTRSM | 1.46 [a] | 1.44 [a] | 59.43 [a] | 19.42 [a] |
| RT | 1.23 [b] | 1.30 [a] | 56.89 [a] | 15.58 [b] |
| FT | 0.93 [c] | 1.35 [a] | 56.71 [a] | 15.36 [b] |
| FTRSM | 1.08 [c] | 1.41 [a] | 59.92 [a] | 19.39 [a] |

FT. flat tillage without any mulch; FTRSM, flat tillage with rice straw mulched; RT. ridge tillage without any mulching; RTRSM, ridge tillage with rice straw mulched. Different lowercase in the same column indicates significant difference between treatments.

### 3.2. Effect of Ridge Tillage and Straw Mulching on the Root Development of Faba Beans

Roots play an important role in absorption and metabolism [30]. The root length density, root surface area, root diameter and root activity of faba beans increased from flowering to harvesting period. The root length density and average diameter increased under RT compared with those under FT at flowering period, but no significant difference in root morphology was observed between RT and FT at harvesting period. Compared with RT, RTRSM significantly increased the surface area at flowering period and root length density, surface area and average diameter at harvesting period. Under FTRSM, root length density was higher at flowering period than at harvest; and root length density, surface area and average diameter were higher at harvesting period than under FT (Table 2). In addition, root activity was the highest under RTRSM at flowering period, followed by RT, FTRSM and FT (Figure 4). The results showed that ridge tillage and straw mulching promote the root growth and activity of faba beans.

**Table 2.** The root morphology of faba bean under different treatments at different periods.

| Treatments | Flowering Period | | | Harvesting Period | | |
|---|---|---|---|---|---|---|
| | Root Length Density (cm cm$^{-3}$) | Surface Area (cm$^2$ cm$^{-3}$) | Average Diameter (mm) | Root Length Density (cm cm$^{-3}$) | Surface Area (cm$^2$ cm$^{-3}$) | Average Diameter (mm) |
| RTRSM | 1.03 [a] | 2.91 [a] | 3.59 [a,b] | 1.7 [a] | 3.82 [a] | 4.96 [a] |
| RT | 0.94 [a] | 1.38 [b] | 3.87 [a] | 1.34 [b,c] | 2.04 [c] | 4.68 [b] |
| FT | 0.82 [b] | 1.17 [b] | 2.75 [b] | 1.22 [c] | 1.98 [c] | 4.71 [b] |
| FTRSM | 0.93 [a] | 1.53 [b] | 3.62 [a,b] | 1.46 [b] | 2.46 [b] | 4.98 [a] |

Different lowercase in the same column indicate significant difference between treatments.

Compared with FT, RT significantly increased the number of root nodules at both flowering and harvesting periods and the dry weight of root nodules at flowering period. Meanwhile, the dry weight of root nodules under RTRSM was higher than that under RT at harvest period (Table 3). The results indicated that ridge tillage increases the number and dry weight of root nodules and straw mulching has minimal effect on the formation of root nodules.

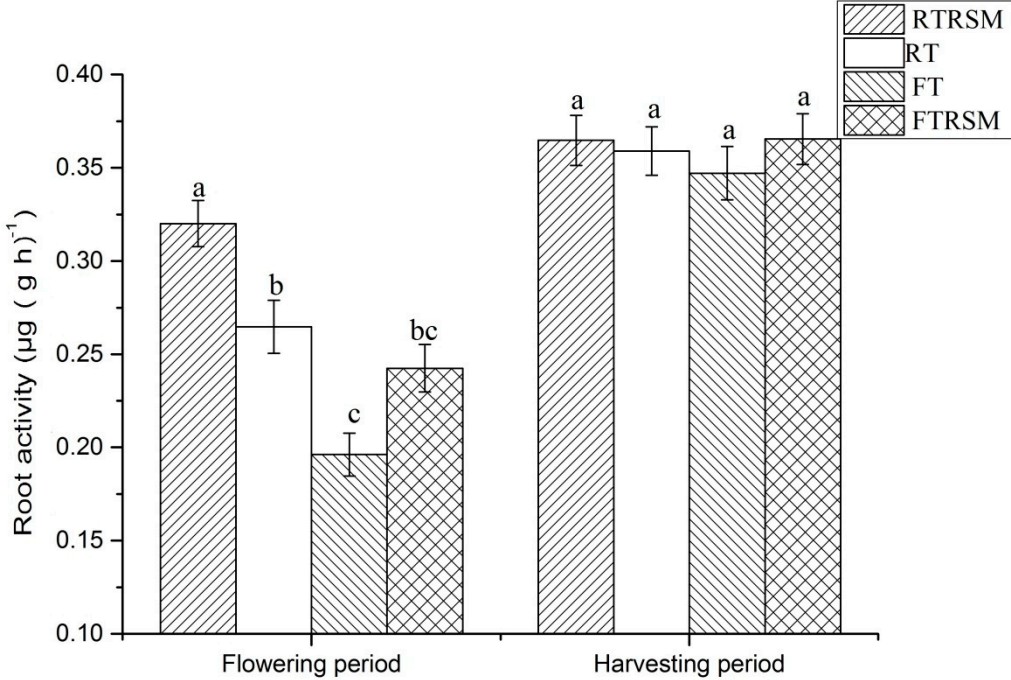

**Figure 4.** The root activity of faba bean under different treatments at different periods. FT. flat tillage without any mulch; FTRSM, flat tillage with rice straw mulched; RT. ridge tillage without any mulching; RTRSM, ridge tillage with rice straw mulched. Different lowercase above histogram indicates significant difference between treatments.

**Table 3.** The root nodules number and dry weight of faba bean under different treatments at different periods.

| Treatment | Root Nodules Number Per Plant | | Root Nodules Dry Weight Per Plant (g) | |
|---|---|---|---|---|
| | Flowering Period | Harvesting Period | Flowering Period | Harvesting Period |
| RTRSM | 56 [a] | 42.3 [a] | 0.207 [a] | 0.151 [a] |
| RT | 54.5 [a] | 41.2 [a] | 0.195 [a] | 0.135 [b] |
| FT | 45.7 [b] | 32.1 [b] | 0.173 [b] | 0.127 [b,c] |
| FTRSM | 48.9 [b] | 34.5 [b] | 0.185 [a,b] | 0.116 [c] |

FT. flat tillage without any mulch; FTRSM, flat tillage with rice straw mulched; RT. ridge tillage without any mulching; RTRSM, ridge tillage with rice straw mulched. Different lowercase in the same column indicates significant difference between treatments.

### 3.3. Effect of Ridge Tillage and Straw Mulching on the Dry Matter and Nutrient Accumulation of Faba Bean

At flowering period, the dry weights of roots, stems and leaves under RTRSM and RT were higher than those under FTRSM and FT. At harvesting period, the dry weights of roots, stems, leaves and seeds under RTRSM and RT were significantly increased compared with those under FTRSM and FT. Meanwhile, the dry matter accumulation in seed under RTRSM was higher than that under RT (Table 4). These results indicated that ridge tillage and straw mulching are beneficial to increase the dry matter accumulation of faba beans.

The nutrient accumulation in roots, stems, leaves and seeds was determined as shown in Table 5. For nitrogen, the accumulation in roots, stems and leaves under RTRSM and RT was higher than that under FTRSM and FT at flowering period. Meanwhile, the accumulation in roots, leaves and seeds under RTRSM and RT was higher than that under FTRSM and FT at harvesting period. Compared with RT, the nitrogen accumulation in the whole plant significantly increased under RTRSM at flowering period. RTRSM and RT increased the phosphorus accumulation in roots, stems, leaves and seeds compared with FTRSM and FT at both flowering and harvesting periods. The phosphorus accumulation

in the whole plant under RTRSM and FTRSM was higher than that under RT and FT at flowering period. The potassium accumulation in roots, stems, leaves and seeds was the highest under RTRSM, followed by RT, FTM and FT at flowering and harvesting periods. The RTRSM significantly increased the potassium accumulation in the whole plant compared with RT at flowering period and FT at harvesting period. These results indicated that ridge tillage and straw mulching improve the accumulation of nitrogen, phosphorus and potassium in faba bean plants.

**Table 4.** The dry matter accumulation of faba bean in different organs under different treatments at different periods.

| Treatment | Flowering Period | | | Harvesting Period | | | |
|---|---|---|---|---|---|---|---|
| | Root (g Plant$^{-1}$) | Stem (g Plant$^{-1}$) | Leaf (g Plant$^{-1}$) | Root (g Plant$^{-1}$) | Stem (g Plant$^{-1}$) | Leaf (g Plant$^{-1}$) | Seed (g Plant$^{-1}$) |
| RTRSM | 5.68 [a] | 31.96 [a] | 11.75 [a] | 7.57 [a] | 36.72 [a,b] | 15.24 [a,b] | 16.33 [b] |
| RT | 5.57 [a,b] | 30.92 [a,b,c] | 11.24 [a,b] | 7.18 [a,b] | 38.86 [a] | 16.75 [a,b] | 18.90 [a] |
| FT | 3.4 [c] | 20.46 [c] | 7.56 [b] | 5.4 [c] | 28.86 [c] | 12.56 [b] | 13.29 [c] |
| FTRSM | 3.60 [c] | 25.62 [b,c] | 9.13 [a,b] | 5.60 [c] | 33.32 [b] | 13.13 [a,b] | 14.06 [c] |

FT. flat tillage without any mulch; FTRSM, flat tillage with rice straw mulched; RT. ridge tillage without any mulching; RTRSM, ridge tillage with rice straw mulched. Different lowercase in the same column indicates significant difference between treatments.

**Table 5.** Nitrogen, phosphorus and potassium accumulation of faba bean in different organs under different treatments at different periods.

| Nutrient | Treatment | Flowering Period | | | | Harvesting Period | | | | |
|---|---|---|---|---|---|---|---|---|---|---|
| | | Root (mg Plant$^{-1}$) | Stem (mg Plant$^{-1}$) | Leaf (mg Plant$^{-1}$) | Total (mg Plant$^{-1}$) | Root (mg Plant$^{-1}$) | Stem (mg Plant$^{-1}$) | Leaf (mg Plant$^{-1}$) | Seed (mg Plant$^{-1}$) | Total (mg Plant$^{-1}$) |
| Nitrogen | RTRSM | 144.52 [a] | 848.44 [a] | 425.92 [a] | 1418.88 [a] | 159.90 [a] | 986.52 [a] | 522.26 [a] | 729.08 [a] | 2397.76 [a] |
| | RT | 115.07 [b] | 729.79 [b] | 383.27 [b] | 1228.13 [b] | 121.52 [b] | 904.78 [b,c] | 505.94 [a,b] | 695.23 [b] | 2227.47 [a] |
| | FT | 65.37 [c] | 681.22 [c] | 269.54 [d] | 1016.13 [c] | 82.66 [d] | 889.56 [c] | 454.84 [c] | 550.74 [c] | 1977.80 [b] |
| | FTRSM | 75.94 [c] | 621.71 [c] | 328.92 [c] | 1026.57 [c] | 105.58 [c] | 833.44 [c] | 490.19 [b] | 585.27 [c] | 2014.48 [b] |
| Phosphorus | RTRSM | 21.55 [a] | 135.92 [a] | 44.64 [a] | 202.12 [a] | 26.18 [a] | 142.32 [a] | 48.92 [a] | 138.64 [a] | 356.06 [a] |
| | RT | 17.91 [b] | 120.74 [b] | 36.63 [b] | 175.27 [b] | 25.45 [a] | 123.02 [b] | 47.8 [a] | 118.26 [b] | 314.53 [a] |
| | FT | 10.25 [c] | 80.65 [d] | 23.92 [c] | 114.82 [d] | 15.66 [b] | 74.71 [d] | 37.93 [b] | 91.59 [c] | 219.89 [b] |
| | FTRSM | 11.90 [c] | 105.00 [c] | 32.86 [b] | 149.76 [c] | 16.88 [b] | 89.91 [c] | 46.01 [a] | 98.35 [c] | 251.15 [b] |
| Potassium | RTRSM | 55.45 [a] | 672.51 [a] | 147.06 [a] | 875.02 [a] | 56.68 [a] | 654.79 [a] | 199.60 [a] | 217.28 [a] | 1128.35 [a] |
| | RT | 47.05 [b] | 579.20 [b] | 135.27 [b] | 761.52 [b] | 48.57 [b] | 571.00 [b] | 173.99 [b] | 205.49 [a] | 999.05 [a,b] |
| | FT | 36.07 [c] | 497.27 [c] | 95.05 [c] | 628.39 [c] | 44.55 [b] | 446.90 [c] | 139.21 [c] | 187.17 [b] | 817.83 [c] |
| | FTRSM | 42.10 [b] | 538.02 [b] | 105.00 [c] | 685.12 [c] | 46.87 [b] | 527.57 [b] | 146.77 [c] | 187.47 [b] | 908.68 [b] |

FT. flat tillage without any mulch; FTRSM, flat tillage with rice straw mulched; RT. ridge tillage without any mulching; RTRSM, ridge tillage with rice straw mulched. Different lowercase in the same column indicates significant difference between treatments.

### 3.4. Effect of Ridge Tillage and Straw Mulching on the Yield Characteristics of Faba Beans

The difference of plant height among the four treatments was not significant. The number of branches in main stem was the largest under RTRSM and the number of nodes in main stem was the largest under RT. RTRSM, RT and FTRSM significantly increased the weight of pod per plant compared with FT. The number of pods per plant under RTRSM and RT was higher than that under FTRSM and FT. Moreover, the 100-seed weight of faba beans under RTRSM was higher than that under RT, FTRSM and FT (Table 6). For the yield of faba beans, RTRSM produced the highest yield, followed by RT, FTRSM and FT. Meanwhile, the yield under FTRSM was significantly increased compared with that under FT (Figure 5). These results indicated that ridge tillage and straw mulching increase the yield of faba beans.

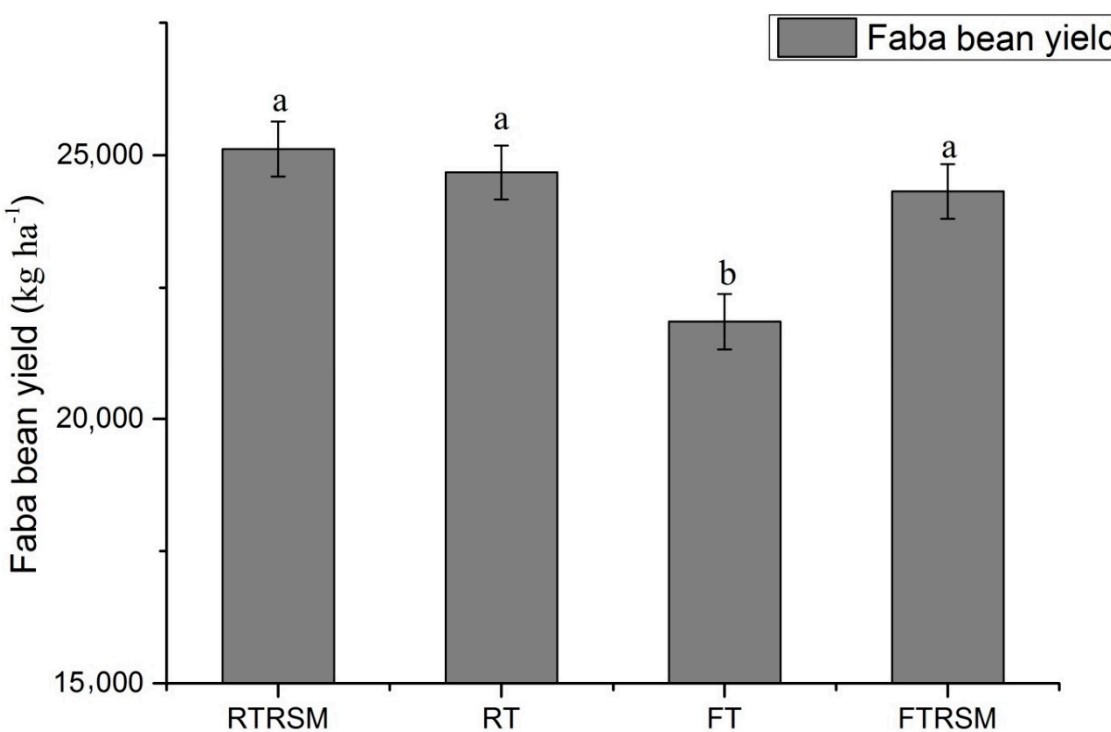

**Figure 5.** The yield of faba bean under different treatments. FT. flat tillage without any mulch; FTRSM, flat tillage with rice straw mulched; RT. ridge tillage without any mulching; RTRSM, ridge tillage with rice straw mulched. Different lowercase above histogram indicates significant difference between treatments.

**Table 6.** The agronomic characteristics of faba bean under different treatments at harvesting period.

| Treatment | Plant Height (cm) | Branches Number of Main Stem | Nodes Number of Main Stem | Pod Weight Per Plant (g) | Pods Number Per Plant | 100-Seeds Weight (g) |
|---|---|---|---|---|---|---|
| RTRSM | 111.17 [a] | 5.67 [a] | 15.50 [b] | 283.04 [a] | 13.25 [a] | 386.9 [a] |
| RT | 108.33 [a] | 5.17 [b,c] | 17.67 [a] | 281.18 [a] | 13.17 [a] | 375.55 [b,c] |
| FT | 112.83 [a] | 5.50 [a,b] | 13.83 [c] | 241.15 [b] | 11.58 [b] | 369.7 [c] |
| FTRSM | 110.83 [a] | 5.63 [a] | 15.33 [b] | 272.31 [a] | 11.92 [a,b] | 381.85 [b] |

FT. flat tillage without any mulch; FTRSM, flat tillage with rice straw mulched; RT. ridge tillage without any mulching; RTRSM, ridge tillage with rice straw mulched. Different lowercase in the same column indicates significant difference between treatments.

## 4. Discussion

In this study, ridge tillage and straw mulching increased the soil temperature and decreased the soil humidity. He et al. [19] reported that relative to conventional tillage, ridge tillage significantly ($p < 0.05$) increases the mean soil temperature to 0.10 m depth by 0.7–2.48 °C in the cold season during the spring maize growing stage. Compared with traditional ploughing, ridge tillage with straw cover increases soil temperature at 0.10 m depth by 1.08 °C in April [31]. Yang et al. reported that ridge tillage generally enhances soil temperature, water content and crop development in Northeast China [32]. The use of ridge-furrow and plastic-mulching tillage can significantly increase the soil temperature [33]. Similar results were reported by Abu-Hamdeh et al. [34]. Radke [35] reported that the application of ridge tillage with residue cover could increase soil temperature due to the change in micro-topography. Ridge tillage increases the area of sunlight absorbed by the soil, thus increasing the soil temperature. However, our results show that ridge and ridge cover straw decrease soil moisture mainly due to the reduction of rain falling into the soil. The study of ridge and straw mulching to improve soil moisture is a long-term result of reducing soil moisture evaporation.

The root length density, root surface area, root diameter and root activity under ridging tillage with rice straw mulched were higher than those under flat tillage. Ridge tillage induces the development of more functional nodal roots compared with no tillage and increases the root length density and surface area of rice [36]. Ridge covered with different mulches materials substantially enhances the rooting systems on the top 50 cm soil profile [37]. Studies in Canada and Australia have also indicated that the root diameter, root length density and root dry weight in legumes and cereals crops are positively correlated to soil water availability [38]. Ridge tillage improves the root growth and lateral root proliferation of corn, during a hot, dry growing season and affects root penetration and distribution [39,40]. Ren et al. reported that ridge tillage could promote plant root growth and development, total root biomass and volume, length and superficial area increase by 14–20%, 14–34%, 9–11% and 14–15%, respectively [41]. A remarkable increase in total root biomass and volume, absorption capacity and ratio of root and shoot was also observed and these effects lay the foundation for acquiring a high yield [42,43]. Qian et al. reported that under ridge tillage, the total absorption area, active absorption area and ratio of root and shoot increase by approximately 21%, 10–17% and 11–40%, respectively [43]. Quan et al. stated that ridge tillage increases the total root biomass per plant by 39–110% compared with FT [44]. Ridge and ridge cover straw also promote root growth and development by providing good soil temperature and moisture.

In this study, ridge tillage increased the nitrogen, phosphorus and potassium absorption of the roots, stems and leaves of faba bean compared with flat tillage. This practice increases the leaf nitrogen, phosphorus and potassium of cocoyam compared with conventional tillage [45]. Compared with conventional flat tillage, ridge tillage substantially increases the accumulation of nitrogen, phosphorus and potassium in the stems and leaves of rice during heading and the accumulation rate of nitrogen, phosphorus and potassium, indicating that ridge tillage could increase the absorption, distribution and accumulation of nitrogen, phosphorus and potassium in the critical period of nutrient accumulation [46]. Ridges enhance the amount of nutrient uptake in the winter wheat/summer maize rotation system [47]. In this work, RT and RTRSM provided good soil temperature and moisture and promoted the nutrient absorption of faba beans. Liu et al. reported that a long-term ridge tillage treatment can stimulate inorganic nitrogen retention capacity and, thus, provides great capacity to supply available nitrogen for uptake by crop plants [48]. Ridge tillage can establish soil functional zones with distinct nitrogen profiles and the relocation of potentially mineralizable nitrogen in-row may increase the spatial efficiency of nitrogen provision relative to conventional tillage [49].

In this study, RTRSM and RT improved soil total nitrogen, total phosphorus, soluble potassium and organic matter compared with FT. Ridge tillage substantially increases total nitrogen, soil microbial biomass nitrogen and soil urease activity in soil aggregates [50]. Ridge tillage variation benefits soil microbial growth, accelerates the decomposition of soil organic matter, promotes the release and absorption of available nutrients and enhances nutrient utilization efficiency [51]. Doraiswamy et al. suggested that soil erosion is controlled and soil carbon sequestration is enhanced with a ridge tillage system [52]. Ridge tillage decreasing soil erosion may be the main reason the higher soil nutrient than that in flat tillage [53]. Straw mulch treatment improves soil total nitrogen, total phosphorus, soluble potassium and organic matter compared with no straw mulch treatment possibly due to the returned crop straw increasing the nitrogen content and activity of urease, phosphatase and invertase in the soil [54].

The yield of RTRSM was increased significantly compared with that of FT. The 3-year average maize yield for ridge tillage was 9.9% higher than that for CT [19]. Li et al. reported that under furrow-ridge mulching cultivation conditions, the grain yield of winter wheat increases by 16–44% [55]. Compared with flat tillage, ridge tillage increases the grain yield of potatoes, summer soybean and soybean by 75–86%, 13–21% and 25%, respectively [56–58]. Tisdall and Hodgson reviewed ridge tillage practice in Australia and stated its successful application, mainly for vegetables or irrigated crops grown on poorly drained alfisols and

vertisols [59]. The better yields from crops grown on ridges/furrow compared with those grown on flat land were attributed mainly to the improved soil moisture conditions, good air permeability and effective soil nutrition supply [30].

## 5. Conclusions

Under RT and RTRSM, the temperature of soil increased, but soil humidity decreased. Both treatments improved soil total nitrogen, phosphorus, available potassium and organic matter content. Ridge tillage and straw mulching changed the soil environment and affected the root growth and nutrient accumulation of faba beans. RT and RTRSM increased the root length density, root surface area, root diameter, root activity and the nitrogen, phosphorus and potassium absorption of roots, stems and leaves of faba beans at flowering and harvest periods. The yield of faba beans was the highest under RTRSM, followed by RT, FTRSM and FT. These results indicated that ridge tillage and straw mulching affect faba bean growth by improving soil moisture conditions and providing good air permeability and effective soil nutrition supply.

**Author Contributions:** F.X. and Y.W. designed the experiments. F.X. guided the research. B.L., X.S. and X.C. performed the experiments, analyzed the data, and wrote the manuscript. X.C., software; B.L., X.S. and X.C, formal analysis; Y.W., resources; B.L., writing—original draft preparation; X.S., Investigation; J.L., F.X. and Y.W., methodology; B.L. and X.C., writing—review and editing; B.L., visualization; J.L., supervision; F.X., project. All authors have read and agreed to the published version of the manuscript.

**Funding:** This study was funded by the National Key Research and Development Program of China (2016YFD030020904) and Jiangsu Province Agricultural Independent Innovation Fund Project [CX (18) 2019].

**Institutional Review Board Statement:** Not applicable.

**Informed Consent Statement:** Not applicable.

**Data Availability Statement:** The data presented in this study are available on request from the corresponding author.

**Conflicts of Interest:** The authors declare no conflict of interest.

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
