# Peer review of "Effects of Ridge Tillage and Straw Mulching on Cultivation the Fresh Faba Beans"

_agronomy, doi:10.3390/agronomy11061054_

Round 1

Reviewer 1 Report

The work was edited contrary to the editorial requirements of the journal.

It is necessary to correct the notation of units. They are to be compatible with the SI system. Why are there old methodologies, no new ones? Improve the description of the methodology, especially the determination of potassium. What is it extracted? I have marked numerous errors in the text. References very badly developed. I have not improved everything because there is a lot. Well written discussion.

Author Response

  1. It is necessary to correct the notation of units.They are to be compatible with the SI system.

Response

----------We have corrected the notation of units in the revised manuscript. (Please see red text in the manuscript).

  1. Why are there old methodologies, no new ones? Improve the description of the methodology, especially the determination of potassium.What is it extracted?

Response

---------- The old methodologies are usually used in some papers and the results are credible [1-5]. We have improved the description of the methodology, especially the determination of potassium (Please see red text in materials and methods section).

[1] Li Y. Environmental contamination and risk assessment of mercury from a historic mercury mine located in southwestern China[J]. Environmental Geochemistry & Health, 2013, 35(1):27-36.

[2] [1] Zhang L H,Zhang S J,Ye G F , et al. Changes of tannin and nutrients during decomposition of branchlets of Casuarina equisetifolia plantation in subtropical coastal areas of China[J]. Plant, Soil and Environment, 2018, 59( 2):74-79.

[3] Lin YM, Liu XW, Zhang H, et al. Nutrient conservation strategies of a mangrove species Rhizophora stylosa under nutrient limitation[J]. Plant & Soil, 2010, 326(s1-2):469-479.

[4] Zhang L ,Liu J, Yi L I . Comparison of the spatial and temporal variability of macroinvertebrate and periphyton-based metrics in a macrophyte-dominated shallow lake[J]. Frontiers of earth science, 2015, 9(1):137-151.

[5] B LI., X.Y. CHEN, X.R. YU, J. LIU,F. XIONG1Returning the Rice Residue Affects Accumulation and Physicochemical Properties of Wheat Starch[J]. Agronomy Journal, 2019, 111(1):39-48.

  1. I have marked numerous errors in the text.References very badly developed.I have not improved everything because there is a lot. Well written discussion.

Response

----------Thanks for reviewer marking numerous errors in the text. We have corrected numerous errors in the text and corrected the References in the revised manuscript. (Please see red text in the manuscript).

The comments and Suggestions in the text‘agronomy-1226036-review’

4.Line 1 propose to add - cultivation the(comments in the text‘agronomy-1226036-review’)

Response

---------- Thanks the reviewer for good suggestion. We have added ‘cultivation the’in the title in the revised manuscript.(Please see red text in the manuscript in line 1).

  1. Line11Text too long. Maksimun 200 words.

Response

---------- Thanks the reviewer for good suggestion. We have simplified the ABSTRACT within 200 words in the revised manuscript.(Please see red text in ABSTRACT section in the manuscript in line 12).

6.Line 32 Chapters should be numbered.

Response

---------- Thanks the reviewer for good suggestion. We have numbered the chapters. (Please see Chapters in the manuscript).

7.Line 33 In the text, reference numbers should be placed in square brackets [ ], and placed before the punctuation; for example [1], [1–3] or [1,3].

It applies to all cited literature.

Response

---------- Thanks the reviewer for good suggestion. We have corrected reference citation in the revised manuscript (Please see red text in the manuscript).

  1. Line37 Is there no newer data?

Response

---------- We have replaced the data in 2016 in the revised manuscript. (Please see red text in line 34 in the manuscript).

  1. Line 76Written contrary to the requirements of the editorial office.

Response

---------- We have writted as the requirements of the editorial office in the revised manuscript. (Please see red text in line 74 in the manuscript).

9.Line 87  It should be written everywhere without a period and -1 in the superscript. kg ha-1. Write the chemical formulas correctly everywhere.

Response

---------- Thanks the reviewer for good suggestion. We have corrected the notation of units and chemical formulas in the revised manuscript. (Please see red text in the manuscript).

10.What is total or soluble potassium?

Response

---------- Soil contains potassium in three forms, readily available potassium (0.1%–2% of total potassium), slowly available potassium (2%–8%), and mineral potassium (90%– 98%)[1]. Available potassium includes water-soluble potassium and exchangeable potassium, which are present in soil as K+[2 ]. The total potassium content of soil is the sum of various forms of potassium, most of which are not readily available to plants.

[1] Prajapati, K., THE IMPORTANCE OF POTASSIUM IN PLANT GROWTH – A REVIEW. Indian Journal of Plant Sciences 2012, 1, 177-186.

[2]Mclean, E. O.; Watson, M. E., Soil measurements of plantavailable potassium. In: Munson RD (ed) Potassium in agriculture 1985, 277–308.

  1. Line99 This needs to be corrected. With which the potassium was extracted.

Response

---------- Thanks the reviewer for good suggestion. We have improved the description of the methodology, especially the determination of potassium (Please see red text in materials and methods section). The water-soluble potassium and exchangeable potassium was extracted.

 12.Line 114 Everywhere, not mL, but cm3

Response

---------- Thanks the reviewer for good suggestion. We have corrected the mL to cm3 in the revised manuscript. (Please see red text in line 199, 121 and 123 in the manuscript).

13.Line 116 It should be 2M and 1M ....

Response

---------- We have corrected the notation of units in the revised manuscript. (Please see red text in line121 in the manuscript).

14.Line 153 It should be like this Figure 1.  Improve everywhere. We put a period at the end of the signature.

Response

---------- Thanks the reviewer for good suggestion. We have corrected the Fig to Figure and puted a period at the end of the signature in the revised manuscript. (Please see Figure 1-5in the manuscript).

15.Line 160 It should be like this Table1.  Improve everywhere. We put a period at the end of the signature.

Response

---------- Thanks the reviewer for good suggestion. We have corrected these and puted a period at the end of the signature in the revised manuscript. (Please see Table1-6in the manuscript).

  1. Line 162 It has to be [g kg-1]. Improve everywhere. We write these designations in superscript. Improve everywhere.

Response

---------- Thanks the reviewer for good suggestion. We have corrected these in the revised manuscript. (Please see Table1 in the manuscript).

17 Line 179 In English, there is no period, only a space.  It should be [cm2 cm-3]

Response

---------- Thanks the reviewer for good suggestion. We have corrected these in the revised manuscript. (Please see Table2 in the manuscript).

18.Line 247 It should be Jin He [number] .......Correct everywhere accordingly. As per the editorial requirements.

Response

---------- Thanks the reviewer for good suggestion. We have corrected these everywhere accordingly in the revised manuscript. (Please see red text in the manuscript).

  1. Line 327 This sentence is unnecessary. This is not a conclusion. This is written earlier in the text.

Response

---------- Thanks the reviewer for good suggestion. We have deleted this sentence in the revised manuscript. (Please see red text in the manuscript).

20.Line330 Soluble or available. These names cannot be used interchangeably.

Response

---------- Sorry for our mistakes and we have corrected this error i the revised manuscript. (Please see red text in line 330in the manuscript).

  1. Line 343They are not written in accordance with the requirements of the journal's editorial office. Lots of the mistakes. There is so much that I am not correcting.

Response

---------- We have checked and rearranged the reference in the revised manuscript. (Please see red text in reference section in the manuscript).

Reviewer 2 Report

The article is well written but with some missing information. Introduction & Materials and Methods section may contain information about tillage or soil management practices (especially those used in this study) supported with a literature review. However, include more detail about the experimental treatments. For example, what is ‘flat tillage’ as it is reported to be the best tillage management practices.

Line 13. Growth of faba bean

Line 13. Give biological name of faba bean too

Line 80. The abstract says that these experiments were conducted during 2016 and 2017

Line 87-88, 97 and elsewhere. Take care of super- and sub-scripts

Line 116. No space between the number and unit of temperature. Take care of subscripts in H2SO4

Line 125 and elsewhere. No space between the number and unit of temperature

Line 142. p < of ≤ 0.05?

Figure 1. Labeling is not readable on axes and bars

Line 154. Was this a statistical significant improvement?

Line 163, 180, 194, 206, 225. Lowercase letters?

Figure 2. Please confirm the unit of root activity on y-axis

Author Response

  1. The article is well written but with some missing information. Introduction & Materials and Methods section may contain information about tillage or soil management practices (especially those used in this study) supported with a literature review. However, include more detail about the experimental treatments. For example, what is ‘flat tillage’ as it is reported to be the best tillage management practices.

Response

----------We have improved information about ridge tillage in Introduction & Materials and Methods section. (Please see red text in Introduction & Materials and Methods section).

  1. Line 13. Growth of faba bean

Response

---------- Thanks for reviewer corrected this error in the text and we have revised it. (Please see red text in line 14)

  1. Line 13. Give biological name of faba bean too

Response

---------- We have given biological name of faba bean in the text. (Please see red text in line 14).

  1. Line 80. The abstract says that these experiments were conducted during 2016 and 2017

Response

---------- Sorry for our mistakes and we have corrected this error in the text. (Please see red text in line 78).

  1. Line 87-88, 97 and elsewhere. Take care of super- and sub-scripts

Response

---------- Sorry for our mistakes and we have revised the chemical formulas in a correct form. (Please see red text in Materials and Methods section).

  1. Line 116. No space between the number and unit of temperature. Take care of subscripts in H2SO4

Response

---------- Sorry for our mistakes and we have corrected this error in the text. (Please see red text in the manuscript).

  1. Line 125 and elsewhere. No space between the number and unit of temperature

Response

---------- There is indeed a blank space between number and unit. But there is no need to insert a blank space between number and the unit of ℃.

  1. Line 142. p < of ≤ 0.05?

Response

---------- We have revised the description to “0.01p<0.05” in statistical analysis section. (Please see red text in line 147)

  1. Figure 1. Labeling is not readable on axes and bars

Response

---------- Sorry for our mistakes and we have revised the label on axes and bars in Figure.

  1. Line 154. Was this a statistical significant improvement?

Response

---------- Thank the reviewer for good suggestion and we have improved the statistical significant.

  1. Line 163, 180, 194, 206, 225. Lowercase letters?

Response

---------- We have used the superscript of lowercase letters to represent significant difference in Tables.

  1. Figure 2. Please confirm the unit of root activity on y-axis

Response

---------- We have confirmed and revised the unit of root activity on y-axis.

Round 2

Reviewer 1 Report

Almost everything is corrected. The remaining four small mistakes.

Line number 119, 120 and 123 - cm3 not cm-3

Table 2 Root length density - (cm cm-3) in one line

Fig 3. - Faba bean not Fababean

Line number 252 - There is no number (19) at He et al 

Author Response

  1. Line number 119, 120 and 123 - cm3not cm-3

Response

----------We have corrected these in the revised manuscript. (Please see the track changes in Line 121, 123 and 125 in the revised manuscript).

  1. Table 2 Root length density - (cm cm-3) in one line

Response

----------We have corrected these in the revised manuscript. (Please see details in Table 2 in the revised manuscript).

3.Fig 3. - Faba bean not Fababean

Response

----------We have corrected the “Fababean” to “Faba bean” in Figure 5. (Please see Figure 5 in the revised manuscript).

4.Line number 252 - There is no number (19) at He et al 

Response

----------We have corrected this in the revised manuscript. (Please see the track changes in Line 254 in the revised manuscript).
